# Improving Anti-PD-1/PD-L1 Therapy for Localized Bladder Cancer

**DOI:** 10.3390/ijms22062800

**Published:** 2021-03-10

**Authors:** Florus C. de Jong, Vera C. Rutten, Tahlita C. M. Zuiverloon, Dan Theodorescu

**Affiliations:** 1Department of Urology, Erasmus University Medical Center, Erasmus MC Cancer Institute, 3015 GD Rotterdam, The Netherlands; f.c.dejong@erasmusmc.nl (F.C.d.J.); v.rutten@erasmusmc.nl (V.C.R.); t.zuiverloon@erasmusmc.nl (T.C.M.Z.); 2Samuel Oschin Comprehensive Cancer Institute, Cedars-Sinai Medical Center, Los Angeles, CA 90048, USA; 3Departments of Surgery (Urology) and Pathology, Cedars-Sinai Medical Center, Los Angeles, CA 90048, USA

**Keywords:** bladder cancer, BCG-unresponsive, neoadjuvant chemotherapy, immune checkpoint inhibition, PD-1, PD-L1

## Abstract

In high-risk non-muscle invasive bladder cancer (HR-NMIBC), patient outcome is negatively affected by lack of response to *Bacillus-Calmette Guérin* (BCG) treatment. Lack of response to cisplatin-based neoadjuvant chemotherapy and cisplatin ineligibility reduces successful treatment outcomes in muscle-invasive bladder cancer (MIBC) patients. The effectiveness of PD-1/PD-L1 immune checkpoint inhibitors (ICI) in metastatic disease has stimulated its evaluation as a treatment option in HR-NMIBC and MIBC patients. However, the observed responses, immune-related adverse events and high costs associated with ICI have provided impetus for the development of methods to improve patient stratification, enhance anti-tumorigenic effects and reduce toxicity. Here, we review the challenges and opportunities offered by PD-1/PD-L1 inhibition in HR-NMIBC and MIBC. We highlight the gaps in the field that need to be addressed to improve patient outcome including biomarkers for response stratification and potentially synergistic combination therapy regimens with PD-1/PD-L1 blockade.

## 1. Introduction

Bladder cancer (BC) is among the ten most common cancers worldwide with 550k new cases yearly and 200k deaths in 2018 [1]. BC is classified into non-muscle invasive (NMIBC) and muscle invasive bladder cancer (MIBC) [2,3]. Based on molecular [3] and clinical-pathological features, a subgroup of NMIBC patients are defined as having high-risk disease (HR-NMIBC), which indicates a high risk of disease recurrence and progression to MIBC [3]. To lower this risk in HR-NMIBC, patients are treated with a transurethral resection followed by intravesical instillations with Bacillus Calmette-Guérin (BCG) immunotherapy [4]. Despite this approach, ~35% of HR-NMIBC patients develop high-grade (HG) recurrences and ~20% develop disease progression to MIBC [5,6]. These so called “BCG-unresponsive” patients are exposed to unnecessary BCG toxicity without any benefit, and for these patients the recommended treatment is surgical removal of the bladder (radical cystectomy (RC)) with a urinary diversion [2,3,7]. Yet some patients have comorbidities that preclude them from a RC, a surgical procedure associated with significant morbidity and mortality, while others have a strong desire for bladder-sparing treatments [2,6,8,9,10]. In addition, as some BCG-unresponsive patients recur without progression to MIBC, a RC may be considered as overtreatment. In summary, both HR-NMIBC and MIBC patients have an urgent need for better treatments.

In recent years, anti-PD-1/anti-PD-L1 immune checkpoint inhibition (ICI) has been proposed as new treatment modality in both BCG-unresponsive HR-NMIBC and MIBC. However, as in metastatic MIBC, clinical response rates to ICI seem unsatisfying. Here, we present a review on the rationale and use of PD-1/PD-L1 inhibition in localized BC. Secondly, we discuss opportunities and challenges on how to improve the response rate to PD-1/PD-L1 ICI using synergistic treatment regimens.

### Evidence Acquisition

A literature search was conducted using ClinicalTrials.gov for ongoing trials on PD-1/PD-L1 combination therapy in localized BC until 1 February 2021. The search strategy was as follows: Bladder Cancer (Condition or Disease) and PD-1 or PD-L1 or any FDA approved PD-1/PD-L1 inhibitors, namely pembrolizumab, atezolizumab, nivolumab, durvalumab, avelumab or cemiplimab (Other Terms). The search resulted in 350 registered trials, six additional trials were identified from other review articles [11,12,13]. After removal of duplicates and terminated trials, 191 trials were assessed for eligibility. Trials with PD-1/PD-L1 ICI combined with chemotherapy, radiotherapy, other immune checkpoint inhibitors or targeted treatments were included. Other non-targeted treatments (e.g., BCG) and studies involving metastatic BC patients were excluded. A total of 37 clinical trials were included in this systematic review (Figure 1). NCT number, population, timing, phase, antibody, additional treatment and primary endpoints were collected for each trial.

## 2. Immune Checkpoint Inhibition

### 2.1. Rationale for Anti-PD-1/PD-L1 Treatment

The checkpoint protein Programmed Cell Death Protein 1 (PD-1) is expressed in activated T lymphocytes and regulates T cell effector functions [14]. PD-1 restrains T cell immune responses [15]. As PD-1 engages with its ligand, programmed death-ligand 1 (PD-L1), T cell exhaustion occurs, thereby downregulating T cell activity—which is currently believed to be a mechanism to prevent auto-immunity [16,17]. The discovery that the PD-1/PD-L1 axis is involved in T cell regulation led to preclinical studies demonstrating that PD-1/PD-L1 is often overexpressed in malignant cells, and several in vivo experiments showed that inhibition of PD-1 caused tumor rejection in mice, whereas induced overexpression of PD-L1 led to tumor colonization and disease progression [15]. In a syngeneic mouse model, with orthotopically implanted MB49 BC cell lines, treatment of mice with a PD-L1 inhibitor (Avelumab) led to a significant tumor reduction compared to BCG alone or a combination regimen [18]. Treatment with BCG and PD-L1 blockade was evaluated in a chemically induced rat bladder tumor model [19]. Interestingly, BCG treatment upregulated PD-L1 expression in tumor tissue, and combined BCG plus anti-PD-L1 therapy demonstrated tumor weight reduction and boosted circulating CD8+ immune responses as compared to BCG alone.

In human samples, several studies showed that PD-1/PD-L1 expression was higher in tumor tissue specimens from BC patients as compared to normal tissue [14,20,21,22]. A meta-analysis in localized bladder cancer found that PD-L1 expression in tumor cells, as assessed by immunohistochemistry (IHC) and different monoclonal antibodies, was correlated to a poorer clinical outcome in BC, but only two studies specifically looked at HR-NMIBC [23]. Interestingly, PD-L1 tumor expression was associated with a higher tumor stage and grade, with abundant expression in BCG-induced granulomata of BCG-unresponsive patients [22]. BCG-unresponsive patients showed upregulation of PD-L1 expression in both tumor and immune cells compared to BCG-responders, and RNA-sequencing revealed baseline PD-L1 to be higher in BCG-unresponsive patients [22,24,25]. Notably, IHC studies did not find a correlation of PD-L1 expression with disease recurrence in BCG-unresponsive patients [25,26]. All in all, the PD-1/PD-L1 pathway seems implicated in the immune response of localized BC.

### 2.2. Biomarkers Predicting Response to Anti-PD-1/PD-L1 Treatment

Based on Phase III studies, ICIs are now included in major urology guidelines for use as second-line treatment in platinum-relapsed patients, and as first-line treatment in platinum-ineligible metastatic bladder cancer patients on the condition of high PD-L1 tumor cell expression [2]. Despite high PD-L1 expression by IHC, many BC patients do not respond to PD-1/PD-L1 inhibitors, so more accurate biomarkers are needed. Thus far, three systematic reviews reported on the use biomarkers in advanced BC and concluded that none of the current biomarkers were of sufficient quality for use in clinical practice [27,28,29].

### 2.3. PD-1/PD-L1 Effectiveness Studies

Several studies are exploring usage of PD-1/PD-L1 monotherapy in localized BC. Currently, no studies have been published on ICI in BCG-unresponsive NMIBC. However, in the Phase-II Keynote-057 study, pembrolizumab monotherapy is being investigated in BCG-unresponsive patients [30]. Interim analyses showed a pathological complete response (pCR) rate at 3 months of 40.6% (*n* = 102), with 18 patients having a durable CR of ≥12 months. The median follow-up was 28.4 months. In patients that underwent a RC due to ICI treatment failure, three had ≥pT2 disease at the time of RC. Thus far, no published data is available on the use and efficacy of biomarkers in the BCG-unresponsive setting.

In MIBC, two Phase II studies, PURE-01 and ABACUS, have published results [12,31,32,33]. In PURE-01, pembrolizumab was used as neoadjuvant ICI, after which a 37% (*n* = 42) pCR rate was observed at RC, whereas 55% (*n* = 63) of patients were downstaged to NMIBC [32]. Overall, 24-month recurrence-free survival (RFS) was 71.7% in 143 patients; RFS based on pathological staging ranged from 95.9% for pCR, 78.8% for localized BC and 39.3% for patients with lymph node disease [34]. High tumor mutational burden (TMB) from pre-pembrolizumab TURBT samples was associated with an increased probability of pCR (*p* = 0.02) in univariate analysis of pre-treatment samples. Post-pembrolizumab TMB was lower compared to baseline TMB (5.0 Mb vs. 10.1 Mb, *p* = 0.005) in 24 matched pre-post treatment samples, suggesting subclonal ICI-resistant tumor expansion [35]. The presence of DNA damage response (DDR) and/or retinoblastoma protein 1 (RB1) gene alterations (52%) were associated with an increased TMB and likelihood of pCR [35]. qPCR analyses of 14 tumor samples of patients without pCR after pembrolizumab revealed upregulation of genes associated with interferon-γ (IFN-γ) and resistance to immune therapy post-treatment compared to baseline [35]. The ABACUS trial reported an overall pCR rate of 31% after treatment with atezolizumab [35]. TMB at baseline was not associated with treatment outcome. Using IHC, patients with pCR demonstrated increased CD8 (*p* = 0.04) and PD-L1 (*p* = 0.21, SP142 levels) and decreased expression of fibroblast activation protein (FAP) compared to patients without pCR (both *p* < 0.01). An 8-gene cytotoxic T cell signature moderately stratified patients for outcome after ICI. A previously developed TGF-β signature was unable to stratify patients.

Overall, PD-1/PD-L1 blockade for localized BC is encouraging, but interpretation of data is hampered by small sample size, a lack of independent validation and patient-derived pre-clinical models for hypothesis testing [27]. Moreover, based on relatively low overall response rates of Keynote-057, PURE-01 and ABACUS, there is clearly room for improvement.

## 3. Opportunities to Improve Efficacy of PD-1/PD-L1 Inhibition

### 3.1. Combined Treatment with Platinum-Based Chemotherapy

Combining PD-1/PD-L1 inhibitors with platinum-based chemotherapy (PBC) may increase tumor immunogenicity [36]. PBC causes DNA damage and induces cell death, thereby attracting antigen presenting cells (APC) [37]. PBC also increases TMB, and tumor-specific neoantigens are presented by MHC-1 and cause cytotoxic T cell activation [38]. While MHC-1 is often downregulated in cancer, in vitro experiments have shown that PBC induces MHC-1 on tumor cells [36,39,40]. IL-12 is essential for antigen presentation; in vivo knockout experiments showed that PBC increases dendritic cell (DC) maturation and leads to an increased ability of DCs to present antigens in an IL-12 dependent manner, resulting in the hypothesis that PBC sensitizes tumors for immune recognition [41]. Experiments in a murine model revealed that T cell costimulatory molecules such as CD80/CD86 are increased in tumor infiltrating immune cells after cisplatin treatment, suggesting that CD80/CD86 expression can be modulated by cisplatin treatment [42]. In vitro experiments showed that PBC induces PD-L1, making PD-L1 an interesting target to inhibit after PBC [39,43,44,45]. PBC may also decrease PD-L2 expression via modulation of the transcriptional regulator STAT6 [46]. As PD-L2 competes with PD-L1 to bind PD-1, decreased expression of PD-L2 after PBC results in enhanced affinity of PD-L1 to PD-1, and increases the relevance of PD-L1 for ICI [47]. The beneficial effects of the addition of PBC to PD-1/PD-L1 blockade is summarized in Figure 2A.

The favorable effects of combined ICI and PBC have been shown in numerous studies [48]. In an MB49 BC subcutaneous model, anti-PD-L1 combined with PBC was more effective than either therapy alone [49]. In an MBT-2 subcutaneous model, anti-PD-L1 combined with PBC was more effective than PBC alone, but counterintuitively, anti-PD-L1 monotherapy was most effective, suggesting that combined treatment results are model-dependent [49]. Immune profiling of baseline tumors by flow cytometry did not demonstrate differences of tumor immune cell populations, and both MB49 and MBT-2 tumors were highly enriched with myeloid-derived suppressor cells (MDSCs). However, expression of the TIGIT checkpoint proteins on immune cells were decreased after PBC treatment in the MB49 vs. MBT-2 model, and the authors suggest that TIGIT may play a role in the diverse anti-tumor responses between models.

Multiple clinical trials evaluate a combination of PBC with PD-1/PD-L1 ICI in MIBC (Table 1) [13]. Neoadjuvant pembrolizumab plus gemcitabine/cisplatin showed a partial response rate to NMIBC at RC of 61% (*n* = 36) (NCT02365766) [50]. In BLASST-1, a multicenter Phase II trial, PBC combination therapy with nivolumab demonstrated a very high pCR rate of 65.8% (*n* = 27) [51]. Gemcitabine combined with pembrolizumab for cisplatin-ineligible patients yielded results comparable to pembrolizumab alone in NCT02365766, a Phase 1b/II study: 45.2% achieved pCR, whereas 51.6% reached <pT1N0 [52]. Interim analysis of SAKK06/17 showed pCR in 30% (*n* = 9) after treatment with gemcitabine and durvalumab pre- and post-operatively, whereas 20% of patients reached non-muscle invasive disease (pT1/pTis) [53]. Preclinical studies and interim study reports are encouraging and suggest a synergistic effect of chemotherapy with PD-L1 blockade. However, pCR rates are a surrogate marker for actual clinical outcome, thus long-term results analyzing disease progression are necessary. More information on treatment sequence and timing of PBC with ICI is also needed before these alternative treatment paradigms can be implemented.

### 3.2. Combined Treatment with Radiotherapy

Ionizing radiation induces double-stranded DNA breaks and produces reactive oxygen species (ROS) leading to cell cycle arrest, apoptosis and cell death [54,55]. Radiation elicits immunogenic effects within the tumor microenvironment (TME) by upregulating and activating the complement pathway and stimulation of IFN-γ release, both implicated in tumor eradication [56,57]. Radiotherapy enhances the adaptive immune system by upregulation of MHC-I, increasing cytotoxic T cell activity. Combining radiotherapy with immune-modulation may inhibit cancer via APC cross-presentation and improvement of T cell priming [58,59,60,61]. By inducing genotoxic stress, radiation prompts NK-mediated innate immune responses [62]. Interestingly, radiation also elicits chemotactic signals that attract MDSCs, instigating T-cell suppression [63,64]. Immunosuppressive effects can also be induced via activation of TGF-β, leading to upregulation of PD-L1 on tumor cells [65]. Combined radiotherapy with PD-L1 inhibition led to synergistic anti-tumor effects in vivo, thus providing the rationale for combined regimens with anti-PD-1/PD-L1 drugs (mechanism depicted in Figure 2A) [66,67,68,69]. Anti-PD-L1 therapy started one day prior and continued until two weeks after local irradiation (12Gy) in an HT1197/MB49 mouse model led to significant tumor growth delay compared to monotherapy [70].

Radiation may have immune-mediated distant effects, termed the abscopal effect, which refers to regression of non-irradiated lesions located elsewhere than the primary radiated site [62]. The abscopal effect is more often detected in mice treated with a combination of radiotherapy and PD-1/PD-L1 than radiation alone [65,71,72]. The underlying mechanism remains unclear, but it is hypothesized that anti-tumor immune responses are amplified. Taken together, radiation therapy stimulates tumor immunogenicity and suggests higher PD-1/PD-L1 susceptibility, hence combined regimes are appealing for investigation in patients with localized BC who prefer a bladder-sparing treatment [59,60,73]. Issues that need to be resolved include determining the optimal fractioning of radiotherapy, timing of ICI and an appropriate balance with regard to side effects associated with each modality [74,75]. Currently, multiple trials investigating radiotherapy and PD-1/PD-L1 blockade in BC are underway but none have been published so far (Table 1) [11].

### 3.3. Combined Treatment with Anti-CTLA-4

For optimal cytotoxic T cell-mediated tumor response, co-stimulation by CD80/CD86 binding to CD28 is required. CD28-CD80/CD86 interaction leads to increased T cell proliferation, differentiation and survival [76]. CTLA-4, a CD28 homolog, is another checkpoint-inhibitor expressed by T cells. CTLA-4 binds competitively with CD80/CD86 on APC’s, resulting in downregulation of T cell proliferation and response. Hence, CTLA-4 blockade can facilitate T cell recognition and increases eradication of malignant cells, thus strengthening the immunological response (illustrated in Figure 2B) [77,78].

In vivo mouse models studying melanoma, ovarian and colon carcinoma showed higher tumor rejection rates and increased tumor induced lymphocyte activity and proliferation after combined therapy with CTLA-4 and PD-1 compared to monotherapy [79,80]. Combining CTLA-4 and PD-L1 inhibitors yields increased CD8+ and CD4+ counts, reduced CD4+ Tregs, increased expression of pro-inflammatory cytokines and a polarization shift from M2 to M1 macrophages [81]. In another study, MB49 BC cells were injected subcutaneously in C57Bl/6 female mice to investigate the response to anti-PD-1 vs. anti-CTLA-4 alone or combined treatment and found that the combination showed increased circulating CD8+ T cells and superior overall survival compared to each monotherapy. In a control group, BCG treatment had no effect on tumor growth [82]. These results were confirmed in a second study that investigated a combination of anti-PD-1/anti-CTLA-4 in an orthotopic mouse model [83].

The NABUCCO-trial recently demonstrated a 46% pCR rate after treatment with ipilimumab (anti-CTLA-4) and nivolumab (anti-PD-1) prior to RC in cisplatin-ineligible stage III BC patients [84]. Of 24 patients, 14 (58%) reached either non-muscle invasive disease or pCR. Biomarker analysis revealed that tumors from pCR had higher frequencies of DDR alterations compared to non-pCR tumors (*p* = 0.03), and trended towards a higher TMB (*p* = 0.056) [84]. An increased TGF-β expression signature was associated with non-response, while response was independent of baseline CD8+ presence or T-effector signatures. However, upregulation of B cell-related genes at baseline correlated with non-response. Non-responding tumors exhibited more stromal B cells than responding tumors (*p* = 0.043) whereas the density of B cells in the tumor compartment was higher (*p* = 0.07) [84]. Overall, NABUCCO indicates that combination regimens with anti-CTLA-4 blockade should be investigated further, as the pCR results are promising. Other studies investigating CTLA-4 plus PD-1/PD-L1 combinations are illustrated in Table 1 and Table 2. No combination regimes have reported data in HR-NMIBC, but considering toxicity profiles, ICI combinations might also prove useful in BCG-unresponsive patients. The extent to which auto-immune symptoms will develop and whether they are manageable will likely determine if combination therapies are feasible and safe [85].

### 3.4. Enhancing Anti-PD-1/PD-L1 Treatment with Targeted Therapies

Poly (ADP-ribose) polymerase (PARP) is involved in DDR. PARP inhibitors are synthetically lethal against tumors with BRCA1/2 mutations, as these cancer cells are the result of genomic instability [86]. Secondly, PARP inhibitor induced genomic instability leads to more somatic mutations and a higher TMB, which in turn may increase the neoantigen load and immunogenicity, which can be exploited using ICI (Figure 2C) [87,88]. In mice, PARP increased expression of PD-L1, while dual blockade of both PARP and PD-L1 may lead to synergistic effects and increased tumor cell death [89]. In a Phase II study in MIBC, combined inhibition of PARP with olaparib and anti-PD-L1 resulted in a systemic immune response demonstrated by release of chemokines, TNF-α, angiogenic factors, tumor infiltrating lymphocytes (TIL) and increased IFN-y expression [90]. Preliminary results showed a pCR of 44.5% (*n* = 28) after treatment with anti-PDL1 and olaparib [91].

TGF-β is overexpressed in many malignancies and has long been implicated as a stimulator of cancer progression by regulation of apoptosis, cell proliferation and initiation of epithelial mesenchymal transition (EMT) [92]. An RNA-based TGF-β signature was associated with non-response to anti-PD-1 therapy in locally advanced BC and anti-PD-L1 therapy in metastatic BC [84,93]. Murine models showed that blocking of TGF-β reduces TGF-β signaling in stromal cells, facilitated T-cell penetration and provoked strong anti-tumor immunity and tumor regression [93]. Interestingly, a bi-functional fusion protein (Bintrafusp-α) targeting anti-PD-L1 and TGF-β has recently been developed, and it is expected to be tested in localized BC [94]. Vascular endothelial growth factor (VEGF) impairs hematopoietic differentiation and like TGF- β causes an immunosuppressive microenvironment in BC, thereby facilitating evasion of the immune response [95]. Anti-VEGF therapy may eliminate immunosuppressive effects, decreases tumor vasculature, and increases perfusion and influx of T cells [96]. Several trials tested anti-VEGF with concurrent chemotherapy, but this did not result in improved clinical outcome for metastatic BC patients [97]. Thus far, no trials have reported on ICI combinations with anti-VEGF in localized BC yet, but in other cancer entities synergistic effects have been identified [98]. The collagen discoidin domain receptor (DDR2), a kinase, is recognized as a metastases promoter [88]. DDR2 depletion increases sensitivity to anti-PD-1 treatment in vivo in BC and multiple other murine models of cancer [99]. Treatment with the anti-tyrosine kinase inhibitor dasatinib (anti-DDR2) and anti-PD-1 resulted in increased CD8+ tumor lymphocytes compared to monotherapy and a reduced tumor load [99]. Thus, a combination treatment of ICI with dasatinib appears to be worth exploring in localized BC. In summary, the synergistic mechanisms of anti-TGF-β, anti-VEGF and anti-DDR2 with PD-1/PD-L1 blockade are depicted in Figure 2D.

Finally, Toll-like receptor (TLR) agonists are being investigated in localized BC. Treatment with BCG in vitro triggers TLRs and prevents deprivation of MHC-II expression, thereby promoting immunogenicity [100,101]. Treatment with the TLR-4 agonist tasquinimod had weak anti-tumorigenic effects in AY-27 rats and MBT-2 mice models [102]. Combination treatment was tested only in mice, but anti-PD-L1 modulated M1/M2 tumor-infiltrating myeloid cells and increased activity and proliferation of CD8+ T-cells were observed, resulting in tumor regression [102]. To summarize, targeted therapies seem promising when combined with anti-PD-1/PD-L1 treatment, though clinical studies are now needed to assess whether these combination strategies truly enhance clinical efficacy of ICI. An overview of currently ongoing combination trials using targeted treatment combinations with PD-1/PD-L1 blockade are included in Table 2.

## 4. Future Directions on PD-1/PD-L1 ICI in Localized BC

The logical next step to improve efficacy of PD-1/PD-L1 inhibition is by addition of conventional treatments such as PBC and radiotherapy. Yet for combination strategies, many questions remain open. For instance, on dosage and timing: do we provide PD-1/PD-L1 blockade only as a neoadjuvant treatment or also as a maintenance treatment? Additionally, do we use treatment combinations concurrently or sequentially? Improved overall response rate in metastatic bladder cancer patients treated with maintenance avelumab after cisplatin are inspiring for such an approach in localized BC [103]. Furthermore, maintenance ICI might be beneficial after NAC or BCG.

Since molecular characterization of BCG-unresponsive patients is evolving, the optimal ICI combinations with targeted treatments have yet to be elucidated. However, for BCG-unresponsive HR-NMIBC patients, it appears that PD-1/PD-L1 ICI with other checkpoint inhibitors, such as CTLA-4 inhibitors, are a promising way forward. For novel checkpoint inhibitors, however, more pre-clinical research should be advocated. It is also worthwhile to explore the efficacy and toxicity of ICI when used intravesically in HR-NMIBC to determine if it maintains the therapeutic benefit seen with systemic administration [104]. There is no doubt that PD-1/PD-L1 blockade is effective in bladder cancer, and improved biomarker-based selection of patients with the potential for maximum benefit will further improve management while minimizing toxicity and financial costs.

## Figures and Tables

**Figure 1 ijms-22-02800-f001:**
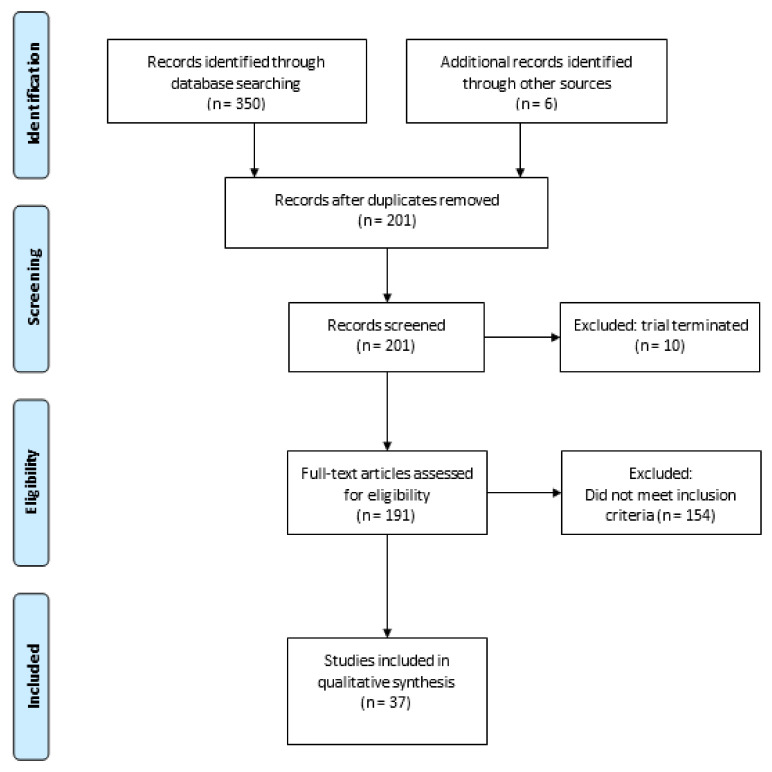
PRISMA flow diagram of evidence acquisition.

**Figure 2 ijms-22-02800-f002:**
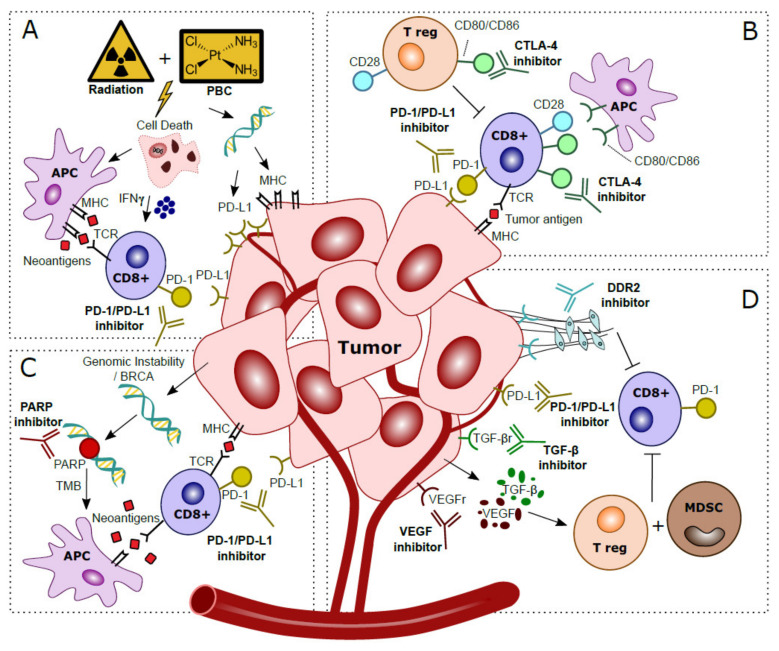
Hypothesized mechanisms of combination treatments to improve clinical response to anti-PD-1/PD-L1 treatments in localized bladder cancer patients. (**A**) Platinum-Based Chemotherapy (PBC) and/or radiation prompts tumor cell death. This process attracts Antigen Presenting Cells (APC), which upregulate presentation of tumor-specific neoantigens to cytotoxic T cells. Activation via IFN-γ released in the tumor microenvironment stimulates anti-tumor immunity. Direct effects of DNA damage by PBC and radiation cause upregulated expression of PD-L1 and MHC in tumor cells. (**B**) CTLA-4 checkpoint inhibitors block CTLA-4 on CD8+ T cells and T regulatory (T reg), thereby further stimulating CD28–CD80/CD86 T cell co-stimulatory responses and anti-tumor immunity. (**C**) Genomic instability is observed in tumor cells, especially in patients with BRCA alterations. The inability to restore DNA damage with blockade of PARP, normally used for DNA repair, leads to more somatic mutations, a higher Tumor Mutational Burden (TMB) and more tumor-specific neoantigens. Neoantigens are presented by Antigen Presenting Cells (APC) to provoke additional cytotoxic T cell responses. (**D**) Blockade of VEGF and TGF-β excreted by the tumor prevents activation of T regulatory cells (T reg) and Myeloid Derived Suppressor Cells (MDSC) that caused an immunosuppressive microenvironment. Inhibition of DDR2 prevents development of Cancer Associated Fibroblasts (CAFs), which stimulated metastases and yielded additional immune suppressive effects and increased CD8+ T cell infiltration.

**Table 1 ijms-22-02800-t001:** Studies investigating anti-PD-1/PD-L1 combined with chemo- and/or radiotherapy in localized bladder cancer.

NCT Number	Population	Timing	Phase	PD-1/PD-L1	Addition	Endpoint(s)
NCT03294304	T2-T4a N0 M0 MIBC	Neoadjuvant	2	Nivolumab	Gem/cis	pRR
NCT02690558	T2-T4a N0 M0 MIBC	Neoadjuvant	2	Pembrolizumab	Gem/cis	pDS to < pT2
NCT03406650	T2-T4a N0-1 M0	Neoadjuvant + adjuvant	2	Durvalumab	Gem/cis	EFS
NCT03661320	T2-T4a N0 M0	Neoadjuvant	3	Nivolumab	Gem/cis	pCRR, EFS
NCT03924856	T2-T4a N0 M0 or T1-T4a N1 M0	Neoadjuvant	3	Pembrolizumab	Gem/cis	pCRR, EFS
NCT03732677	T2-T4a N0-1 M0	Neoadjuvant + adjuvant	3	Durvalumab	Gem/cis	pCRR, EFS
NCT03558087	T2-T4a N0 M0	Neoadjuvant	2	Nivolumab	Gem/cis	CRR, pCRR
NCT04099589	T2-T4a N0 M0	Neoadjuvant	2	Toripalimab	Gem or cis	pCRR
NCT03674424	T2-T4a Nx M0	Neoadjuvant	2	Avelumab	Gem/cis or ddMVAC or paclitaxel + gemcitabine	pCRR
NCT02365766	T2-T4a N0 M0	Neoadjuvant	1/2	Pembrolizumab	Gem/cis or gemcitabine	AE, PalR
NCT04164082	BCG-unresponsive	Neoadjuvant	2	Pembrolizumab	Gemcitabine hydrochloride	CRR, EFS
NCT02560636	T2-T4a N0-3 M0-1	Neoadjuvant	1	Pembrolizumab	Radiotherapy	MTD, AE
NCT02891161	T2-T4a N0-2 M0	Neoadjuvant	2	Durvalumab	Radiotherapy	DLT, PFS, DCR
NCT03950362	BCG-unresponsive	Ineligible/refusal of RC	2	Avelumab	Radiotherapy	RFS
NCT03317158	BCG-unresponsive	Neoadjuvant	1/2	Durvalumab	EBRT + BCG	Recommended dose, RFS
NCT03775265	T2-4a N0 M0	Bladder sparing	3	Atezolizumab	Gem or CIS or fluorouracil + MMC + radiotherapy	EFS
NCT02662062	T2-T4a Nx M0	Ineligible/refusal of RC	2	Pembrolizumab	Cisplatin + radiotherapy	AE (grade 3–4)
NCT02621151	T2-T4a N0 M0	Ineligible/refusal of RC	2	Pembrolizumab	Gemcitabine + radiotherapy	DFS
NCT03617913	T2-T4a N0 M0	Neoadjuvant	2	Avelumab	Flourouracil + MMC or cisplatin and radiotherapy	CRR
NCT03702179	T2-T4a	Neoadjuvant	2	Durvalumab	Tremelimumab + radiotherapy	pRR
NCT03601455	T2-T4a or N+/M+	Ineligible for RC	2	Durvalumab	Tremelimumab + EBRT	AE, PFS
NCT03549715	T2-T4a N0-1 M0	Neoadjuvant	1/2	Durvalumab	Tremelimumab + ddMVAC	AE, pCRR

AE = adverse events, DCR = disease control rate, ddMVAC = dose dense methotrexate vinblastine doxorubicin cisplatin, DFS = disease-free survival, DLT = dose limiting toxicity, EFS = event-free survival, EBRT = External Beam Radiotherapy, Gem/cis = gemcitabine/cisplatin, MMC = mitomycine C, MTD = maximum tolerated dose, PalR = pathologic muscle invasive response rate, p(C)RR = pathological (complete) response rate, pDS = pathological downstaging, PFS = progression-free survival, RFS = recurrence-free survival.

**Table 2 ijms-22-02800-t002:** Studies investigating anti-PD-1/PD-L1 combined with immune checkpoint inhibitors and targeted treatments.

NCT Number	Population	Timing	Phase	PD-1/PD-L1 Antibody	Additional Treatment	Primary Endpoint(s)
NCT03472274	T2-T4a N0-1 M0	Neoadjuvant	2	Durvalumab	Tremelimumab	pCRR
NCT02812420	T2-T3a N0 M0	Neoadjuvant	1	Durvalumab	Tremelimumab	AE
NCT03520491	T2-T4a N0 M0	Neoadjuvant	2	Nivolumab	Ipilimumab	Patients proceeding to RC
NCT03387761	T3-T4a N0 M0 or T1-4a N1-3 M0	Neoadjuvant	1	Nivolumab	Ipilimumab	Patients proceeding to RC
NCT04430036	T2-T4a N0-1 M0	Neoadjuvant	2	Balstilimab	Zalifrelimab + gem/cis	pDS to pT0
NCT04579133	T2-T4a N0 M0 or T1-T4a N1 M0	Neoadjuvant	2	Durvalumab	Olaparib	pCRR
NCT03534492	T2-T4a N0 M0	Neoadjuvant	2	Durvalumab	Olaparib	pCRR
NCT03832673	T2-T3a N0 M0 TCC	Neoadjuvant	2	Pembrolizumab	Epacadostat	pCRR
NCT02845323	T2-T4a N0-N2 M0	Neoadjuvant	2	Nivolumab	Urelumab	CD8 density
NCT04209114	T2-T4a N0 M0	Neoadjuvant + adjuvant	3	Nivolumab	Bempeg	pCRR, EFS
NCT03924895	T2-T4a N0 M0 or T1-T4aN1M0	Neoadjuvant	3	Pembrolizumab	Enfortumab vedotin	pCRR, EFS
NCT03258593	BCG-unresponsive	Neoadjuvant	1	Durvalumab	Vicinium	AE
NCT03978624	T2-T4a N0-x M0	Neoadjuvant	2	Pembrolizumab	Entinostat	Change in CD8 immune signature
NCT03773666	T2-T4a N0 M0	Neoadjuvant	1	Durvalumab	Oleclumab	Patients receiving a single dose followed by surgery
NCT03532451	T2-T4a N0-1 M0	Neoadjuvant	1B	Nivolumab	Lirilumab	AE

AE = adverse events, DFS = disease free survival, DLT = dose limiting toxicity, EFS = event-free survival, pCRR = pathological complete response rate, RR = response rate.

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
