# Peer review of "Improving Anti-PD-1/PD-L1 Therapy for Localized Bladder Cancer"

_ijms, 2021, doi:10.3390/ijms22062800_

Round 1
Reviewer 1 Report
The work is interesting and deserves consideration but the manuscript lacks the "materials and methods" section. As a review, it is necessary to follow the PRISMA criteria as required by the journal itself (the manuscript should use the same structure as research articles, as reported in submission guidelines). Moreover, the reader can not easily understand if the work is a narrative review, systematic review or meta-analysis
Author Response
We addressed the review criteria by implementing PRISMA 2009 Flow Diagram and other issues.
Reviewer 2 Report
Aim of the manuscript was to describe challenges and opportunities of PD-1/PD-L1 inhibition in HR-NMIBC. Manuscript is interesting and it addresses a still debated topic in Literature. Quality of the manuscript and studies reported are good and review well performed.
Author Response

(The authors gave the same response as above.)

Round 2
Reviewer 1 Report
The authors improved the manuscript as requested